# Endoscopic Management of Esophageal Cancer

**DOI:** 10.3390/cancers14153583

**Published:** 2022-07-22

**Authors:** Christopher Paiji, Alireza Sedarat

**Affiliations:** Vatche and Tamar Manoukian Division of Digestive Diseases, David Geffen School of Medicine at UCLA, Los Angeles, CA 90095, USA; cpaiji@mednet.ucla.edu

**Keywords:** staging, diagnosis, esophageal cancer, endoscopic surgery, oncologic outcomes

## Abstract

**Simple Summary:**

Esophageal cancer is a leading cause of cancer-related mortality worldwide. With ongoing innovation in endoscopy and improved understanding of this disease, minimally-invasive endoluminal procedures have served a growing role in diagnosis, treatment, and palliation. In this review article, we discuss the current indications and techniques of endoscopic management of esophageal cancer.

**Abstract:**

Advances in technology and improved understanding of the pathobiology of esophageal cancer have allowed endoscopy to serve a growing role in the management of this disease. Precursor lesions can be detected using enhanced diagnostic modalities and eradicated with ablation therapy. Furthermore, evolution in endoscopic resection has provided larger specimens for improved diagnostic accuracy and offer potential for cure of early esophageal cancer. In patients with advanced esophageal cancer, endoluminal therapy can improve symptom burden and provide therapeutic options for complications such as leaks, perforations, and fistulas. The purpose of this review article is to highlight the role of endoscopy in the diagnosis, treatment, and palliation of esophageal cancer.

## 1. Introduction

There are about 500,000 incident cases of esophageal cancer per year worldwide [1]. While esophageal squamous cell carcinoma (ESCC) is the most common histological subtype globally, esophageal adenocarcinoma (EAC) is more prevalent in the United States and other Western countries, with the incidence noted to be rising in Eastern countries as well [2].

The average 5-year survival rate of esophageal cancer is less than 25% [3]. This poor prognosis has been attributed to late diagnosis, with reportedly 40% of esophageal cancers in the USA diagnosed with distant metastases and another 32% diagnosed with local organ and lymph node involvement [4]. A number of hereditary and environmental risk factors have been identified for both ESCC and EAC, which support different underlying pathobiologies between the two subtypes [5].

Endoscopy has been essential in the detection of lesions suspicious for esophageal cancer and tissue acquisition to make an accurate histologic diagnosis [6]. As our understanding improves regarding precancerous lesions such as Barrett’s esophagus (BE), a known precursor of EAC, endoscopy has promoted early detection, surveillance, and prevention of progression [7]. Furthermore, advances in technology have increased the therapeutic applications of endoscopy in curative resection of early cancer and palliation of advanced malignancy. In this article, we discuss the current indications and techniques of endoscopic management of esophageal cancer.

## 2. Evaluation of Barrett’s Esophagus

Barrett’s esophagus (BE) represents metaplastic columnar epithelium with goblet cells replacing the normal stratified squamous epithelium lining the esophagus [8]. It has been reported to be present in about 5–15% of patients with gastroesophageal reflux disease (GERD) [9]. BE is a precursor to development of esophageal adenocarcinoma (EAC), which carries a 5-year survival rate of <15% in Western societies [10,11]. This necessitates efforts to screen the population and intervene in patients who are at higher risk of developing BE neoplasia.

### 2.1. Indications for Screening

Through histologic and molecular events, BE can progress through stages that include intestinal metaplasia (IM), low-grade dysplasia (LGD), high-grade dysplasia (HGD), intramucosal carcinoma, and ultimately to EAC [12]. Given the emergence of various therapies to treat dysplastic BE and early EAC, at-risk patients would benefit from screening [13]. However, determining who would benefit from endoscopic screening has been a matter of debate.

A large meta-analysis showed that prevalence of BE was 0.8% in a low-risk population [14]. The prevalence was significantly higher in populations with one or more risk factors, including 23.4% among patients with a family history of BE. Furthermore, a meta-regression as part of this analysis demonstrated a linear relationship between the number of risk factors and the prevalence of BE.

Multiple professional societies in the field of gastroenterology have authored guidelines pertaining to criteria for screening of patients. The American College of Gastroenterology (ACG) suggests a single screening endoscopy be performed in patients with chronic GERD symptoms and at least 3 or more additional risk factors: male sex, age greater than 50 years, White race, tobacco smoking, obesity, and family history of BE [15]. On the other hand, the American Society for Gastrointestinal Endoscopy (ASGE) recommends screening in all patients with a family history of EAC or BE, in addition to patients with GERD and at least one additional BE risk factor [16]. Lastly, the European Society of Gastrointestinal Endoscopy (ESGE) recommends consideration of screening in patients with both GERD symptoms for over 5 years and multiple other risk factors [17].

### 2.2. Methods of Screening

Recent guidelines support thorough evaluation of BE using white-light endoscopy (WLE), and newer-generation high-definition endoscopes have allowed for better visualization of esophageal mucosa to improve detection of dysplasia and carcinoma [18,19]. Magnification endoscopes also allow endoscopists to vary the degree of magnification and examine mucosa much closer to the tip of endoscope without losing focus [20,21]. In order to identify any lesions that may be missed by WLE, it has been recommended that patients undergo evaluation via dye-based or electronic chromoendoscopy.

Acetic acid chromoendoscopy involves the use of acetic acid spray, which reversibly alters proteins in cells and causes an acetowhite reaction in the esophagus [22]. This can aid evaluation of the surface pattern of IM, and the observation that dysplastic tissue will lose whitening more rapidly than nondysplastic tissue further augments diagnostic efforts in BE. This technique has been previously studied in combination with magnification endoscopy in a randomized controlled trial and has found to significantly improve the diagnostic yield of tissue acquisition for BE compared to random biopsies [23].

With the technique of Lugol chromoendoscopy, the esophageal surface is sprayed with Lugol’s iodine solution [24]. The glycogen-containing squamous epithelium is stained black, brown, or dark green with absorption of this solution [25]. This facilitates detection of abnormal squamous epithelium, as evidenced by absent dye uptake, particularly in patients with squamous dysplasia or IM [26]. Lugol’s iodine has shown to be effective in delineating squamous mucosa from columnar mucosa in BE [27]. Other dyes that have been studied for use in chromoendoscopy include methylene blue and indigo carmine [28].

Electronic (or virtual) chromoendoscopy involves advanced endoscopic imaging techniques that allow for detailed contrast enhancement of mucosal and vascular surface patterns [29]. This technology includes narrow banding imaging (NBI), which utilizes spectral optical filters to produce two narrow bands of light that can highlight the blood vessels of the mucosa [30]. The use of NBI has demonstrated the potential to improve the detection of Barrett’s esophagus and dysplasia [31]. One prior blinded, tandem study found that higher grades of dysplasia were detected by NBI compared to WLE, and lower mean number of targeted biopsies were obtained with NBI compared to random biopsies with WLE [32].

Reported advantages of using electronic chromoendoscopy over dye-based alternatives include the convenience (ability to switch back to WLE from NBI), shorter procedure time, and preventing the need to assemble and administer dye [33,34]. Novel modalities of optical enhancement, contrast, virtual chromoendoscopy, and artificial intelligence have continued to develop and are slowly entering clinical practice.

### 2.3. Diagnosis of Barrett’s Esophagus

ACG recommends the diagnosis of BE should require the presence of metaplastic columnar epithelium of at least 1 cm in length in the esophagus [15]. Use of this 1 cm cutoff is supported by prior studies highlighting the extremely low risk of progression to dysplasia and EAC with segments less than 1 cm [35,36]. To provide guidance on the endoscopic classification of BE, the Prague C&M criteria were developed by an international working group in 2006 [37]. These criteria involve documenting measurements pertaining to the circumferential extent (C value) and maximum extent (M value) of BE, which was found to have high overall validity. Furthermore, the diagnosis of long-segment BE has been defined by the presence of metaplastic columnar epithelium of at least 3 cm in length [38]. After endoscopic identification of BE, it is recommended that 4-quadrant biopsy sampling is performed every 2 cm in patients without dysplasia and every 1 cm in patients with history of dysplasia (Seattle protocol) [39].

## 3. Management of Barrett’s Esophagus

### 3.1. Nondysplastic Barrett’s Esophagus

Patients with non-dysplastic BE are recommended to undergo endoscopic surveillance in 3–5 years [15]. A meta-analysis demonstrated lower EAC-related and all-cause mortality associated with regular surveillance (relative risk, 0.60; hazard ratio, 0.75) [40]. However, there are no randomized controlled trials that demonstrate improvement in mortality with endoscopic surveillance of BE. Given prior studies have demonstrated length of BE to be a predictor of progression, ACG recommends endoscopy every 5 years for short segment BE and every 3 years for long segment BE [15,41,42]. Specifically, the annual incidence of EAC has been reported to be 0.3–0.6%, and annual combined incidence of HGD and EAC to be 0.9–1.0% [42]. In addition to endoscopic surveillance, it is recommended that patients start a low-dose proton pump inhibitor (PPI), as this has been associated with decreased risk of HGD and EAC [43].

### 3.2. Barrett’s Esophagus with Indefinite Dysplasia

The diagnosis of BE with indefinite dysplasia (IND) is made when pathologic features are identified that that may overlap with dysplasia but are not sufficient for the diagnosis of dysplasia and may be related to active inflammation [44,45]. A prior meta-analysis found that among patients with IND, pooled incidence of HGD and/or EAC was 1.5 per 100 person-years and pooled incidence of EAC was 0.6 per 100 person-years, which is similar to previously reported rates of progression in patients with LGD [46,47]. As with diagnosed cases of dysplasia, it is recommended that diagnosis of IND be confirmed with an expert gastrointestinal pathologist [48]. ACG advises these patients are treated with intensified medical antireflux therapy to heal any underlying reflux esophagitis and undergo repeat endoscopy within 6 months [15]. If repeat endoscopy demonstrates regression to nondysplastic BE or progress to LGD, patients should undergo surveillance according to those algorithms. If repeat endoscopy demonstrates IND, patients should continue surveillance endoscopy every 12 months.

### 3.3. Barrett’s Esophagus with Low-Grade Dysplasia

Diagnosis of LGD presents a challenge, as prior studies have shown high interobserver variability among pathologists [49,50]. It is thus recommended that cases diagnosed with LGD are confirmed with a second experienced pathologist, as cases of confirmed LGD have reported annual progression rates of 9.1–13.4% compared to annual progression rates of 0.49–0.6% in cases downgraded to nondysplastic BE [51,52].

Prior studies have shown the effectiveness of endoscopic eradication therapy (EET) compared to surveillance in preventing progression of BE. The Surveillance vs. Radiofrequency Ablation (SURF) multicenter randomized trial found ablation reduced risk of progression to HGD/EAC by 25.0% and led to complete eradication of IM in 88.2% of patients over 3 years of follow-up [53]. Furthermore, a large meta-analysis demonstrated a significant reduction in risk (RR 0.14, 95% CI: 0.04–0.45, *p* = 0.001) of disease progression in patients who underwent ablation [54]. However, given the risk of adverse events of ablation therapy and the likelihood that neoplastic progression detected on surveillance will be amenable to endoscopic therapy, a shared decision-making approach between patients and physicians is recommended to determine management of LGD [55,56]. If a decision is made to pursue surveillance only, ACG recommends repeat endoscopy every 6 months for one year followed by annual surveillance [15].

### 3.4. Barrett’s Esophagus with High-Grade Dysplasia

Patients with HGD have an annual risk of progression of 6–19% per year [57]. Once confirmed by a second experienced pathologist, it is generally advised that patients with HGD are treated with EET instead of undergoing surveillance alone [15,56,58]. In patients with flat HGD, AGA recommends repeat endoscopy within 6–8 weeks, citing the majority of patients with HGD having a visible lesion [58]. All visible lesions should first be resected, serving an important tool for accurate diagnosis in addition to therapy. Following this, it is advised patients undergo EET of remaining BE [15]. When comparing EET vs. esophagectomy for treatment of HGD/intramucosal EAC, prior systematic review and meta-analysis found no difference with regards to complete eradication, overall survival, and EAC-related mortality [59]. Given the morbidity rate associated with esophagectomy, it is advised that patients are treated with HGD with EET over esophagectomy [15,56,58]. Following EET and achievement of complete eradication of intestinal metaplasia (CEIM), ACG recommends patients undergo surveillance at 3, 6, and 12 months, followed by annual surveillance [15].

## 4. Evaluation of Esophageal Squamous Cell Carcinoma

Development of ESCC occurs in a stepwise process that starts with low-grade intraepithelial neoplasia, followed by high-grade intraepithelial neoplasia and invasive carcinoma [60]. Early detection of ESCC has shown to be associated with favorable outcomes pertaining to successful resection and lower rates of lymph node involvement [61].

There are no screening guidelines for ESCC in the United States given its relatively lower incidence compared to EAC [62]. However, in endemic regions and high-risk populations, screening programs have been found to provide mortality reduction while remaining cost effective [63,64]. Populations that may benefit from screening include those with the following risk factors: smoking, alcohol use, male gender, prior caustic ingestion, prior head and neck cancer, consumption of foods rich in nitrogenous components and areca nuts, nutritional deficiencies, and genetic conditions including tylosis [65,66,67].

Lugol chromoendoscopy is the most widely accepted technique for endoscopic evaluation of ESCC, as early lesions may be missed by using WLE alone [62,68]. As previously discussed, Lugol chromoendoscopy can help identify abnormal squamous epithelium in patients with squamous dysplasia. In a prospective study, 55% of patients with moderate squamous dysplasia and 23% with severe squamous dysplasia were identified only after use of this staining [26].

Additional endoscopic techniques for diagnosis of ESCC have been studied as adjuncts to Lugol’s chromoendoscopy in order to improve diagnostic accuracy. Confocal laser endomicroscopy (CLE) allows for sufficient magnification to provide views of the cells of esophageal squamous epithelium and vascular networks [69]. This technique is associated with high rates of diagnostic accuracy of early ESCC and significant degree of interobserver agreement [70]. High-resolution microendoscopy (HRM) has been introduced as an alternative that is significantly lower in cost than CLE [71]. After application of a topical fluorescent agent, HRM utilizes a fiberoptic microendoscope probe to depict cellular features. A prospective trial found that HRM significantly improved positive predictive value and specificity in the evaluation of ESCC, and this technique could also reduce unnecessary biopsies.

## 5. Staging of Esophageal Cancer

Once histologic cancer diagnosis has been made, accurate staging is important to determine treatment options for patients in order to optimize outcomes [72]. The eighth edition of the American Joint Committee on Cancer (AJCC)/Union for International Cancer Control (UICC) staging manuals for esophageal cancer was released in 2017, featuring subcategorization of pT1 cancer as pT1a and pT1b in addition to a simplified esophagus-specific regional lymph node map [73]. It is recommended that staging begins with a contrast-enhanced computed tomographic (CT) scan of the chest and abdomen and/or fluorodeoxyglucose-positron emission tomography (PET-CT) scan to evaluate for metastatic disease [72]. If distant metastatic disease is not found on CT or PET-CT, endoscopic ultrasound (EUS) should be performed for locoregional staging.

EUS utilizes a combination of both endoscopy and ultrasonography through high-frequency sound waves to evaluate the esophageal wall layers and regional lymph nodes [74]. This can help stage superficial esophageal cancers that include Tis (malignant cells confined by the basement membrane), T1a (extension to lamina propria or muscularis mucosa), and T1b (invasion into submucosa) [73]. Differentiating between T1a and T1b cancers has significant implications on treatment options and prognosis, as a prior study of the National Cancer Data Base found 17% of patients with T1b cancers had lymph node metastases compared to 5% of patients with T1a cancers [75]. Prior meta-analyses have reported EUS had pooled sensitivities and specificities of 84–85% and 87–91%, respectively, for T1a staging [76,77]. For T1b staging, pooled sensitivities and specificities were reported to be 83–86% and 86–89%, respectively.

EUS also has a role in accurate staging of more advanced cancer, including T2 (invasion into muscularis propria), T3 (invasion into adventitia), and T4 (invasion into adjacent structures) [73]. For T3 staging, a prior study found EUS had a sensitivity of 100% and specificity of 83% [78]. Furthermore, a meta-analysis had reported EUS had a pooled sensitivity and specificity of 92.4% and 97.4%, respectively, in T4 staging [79].

Lastly, this endoscopic imaging technique is helpful for evaluation of lymph node involvement, another essential aspect of staging. One prior study has reported features of malignant lymph nodes to include hypoechoic pattern, width of 10 mm or greater, round shape, and sharp borders, with an accuracy of 80% if all four features are present [80]. Furthermore, EUS with fine needle aspiration (EUS-FNA) can be performed for cytologic confirmation of metastatic disease [81]. A prospective study found that accuracy was 87% for EUS-FNA compared to 74% for EUS alone [82]. Overall, the utilization of EUS has been associated with increased likelihood of patients receiving cancer treatment and improved 1-year, 3-year, and 5-year survival [83].

NBI is another endoscopic modality that can assist with T staging through enhancing visualization of microvascular structures, which corresponds with depth of cancer invasion in ESCC [84]. Prior studies in the assessment of superficial squamous cell cancers of the head and neck have found that early lesions have a brown and well-demarcated appearance, and this understanding has been applied to assessment of ESCC [85]. NBI has shown to increase accuracy for diagnosis of the depth of invasion compared to magnifying endoscopy alone [86].

## 6. Endoscopic Resection of Early Adenocarcinoma and Squamous Cell Carcinoma

Endoscopic resection (ER) of visible lesions in BE can provide larger histology specimens, which is important for accurate diagnosis of dysplastic BE. In a multicenter study, endoscopic mucosal resection (EMR) specimens were found to have higher interobserver agreement on the diagnosis of dysplasia compared to biopsy specimens, owing to the finding that vast majority of EMR specimens had submucosa present [87]. In addition, two studies evaluated patients who underwent ER of Barrett’s neoplasia and found that this led to upstage/downstage of histologic grade in 30–49% of patients [88,89].

ER has also shown to improve diagnostic efforts in early esophageal cancer. It has been previously reported that ER can accurately confirm depth of tumor invasion to differentiate between mucosal and submucosal carcinoma, and negative lateral and deep margins can predict the lack of residual tumor following esophagectomy [90]. Furthermore, ER has allowed for assessment of other important prognostic variables including grade of differentiation and presence of lymphovascular invasion [91]. Thus, ER potentially provides valuable information for tumor staging, which can add to other diagnostic modalities including EUS to accurately stage superficial esophageal cancer [72].

### 6.1. Endoscopic Mucosal Resection

#### 6.1.1. Candidates for Endoscopic Mucosal Resection

For T1 esophageal cancer, pathological subclassification subtypes have been defined for mucosal and submucosal involvement [92]. With regards to mucosal involvement, the three types include cancer limited to mucosal epithelium (M1), invasion into lamina propria (M2), and invasion into but not through muscularis mucosa (M3). Submucosal involvement is subclassified to invasion within the shallowest one-third portion of submucosal layer (SM1), intermediate one-third of submucosal layer (SM2), and deepest one-third portion of submucosal layer (SM3).

In patients with adenocarcinoma, EMR can be utilized for curative resection of mucosal (M1-3) lesions without lymphovascular invasion, as multiple studies have shown the very low risk of lymph node metastasis [93]. A single center study found that in patients with BE with suspected HGD or intramucosal carcinoma, EMR led to eradication of neoplasia and BE in 98.8% of patients who completed therapy per-protocol, with most common complication being strictures (41.5%) that were managed with dilation [94].

In patients with submucosal involvement, it has been recommended to avoid EMR given the higher risk of nodal involvement and residual disease [95]. Furthermore, in patients with ESCC, curative resection can be achieved with M1-2 lesions and should be considered only in selected cases of M3 lesions due to higher reported lymph node metastases (11.8%) in ESCC compared to in EAC [96].

#### 6.1.2. Injection-Assisted EMR

There are multiple techniques that can be utilized to achieve resection of neoplasia using EMR. Injection-assisted EMR involves injection of a solution in the submucosal space to lift the lesion for capture by snare [97]. This method helps minimize injury to the deeper layers of tissue. Over the years, various agents including hyaluronic acid, hydroxypropyl methylcellulose, succinylated gelatin, and other synthetic agents have been introduced to promote longer lasting submucosal cushions to facilitate resection [98].

#### 6.1.3. Ligation-Assisted EMR

In ligation-assisted EMR, the target lesion is suctioned into the banding cap followed by deployment of band to create a pseudopolyp (Figure 1) [99]. The band ligation device allows for insertion of an electrocautery snare to resect the lesion above or below the band. One prior study found that this technique was associated with complete endoscopic resection in 92.3% of patients with squamous intraepithelial neoplasia, and complications included acute bleeding (7.6%) and esophageal stricture (1.9%) [100].

In cap-assisted EMR, the lesion is first lifted using a submucosal injection, and then a cap is preloaded onto the tip of the endoscope [101]. A specially designed snare is opened and positioned appropriately within the cap. The lesion is suctioned into the cap, followed by snare closure and resection by using electrocautery. This technique has achieved eradication of neoplasia in 91% of patients but also with stricture rates of 40% [101].

### 6.2. Endoscopic Submucosal Dissection

Endoscopic submucosal dissection (ESD) involves the resection of tumors through dissection of the submucosal plane [93]. The potential advantage of this technique over EMR is the ability for ESD to remove tumors en bloc regardless of size, which may improve precision of pathologic staging and lower recurrence rate [102].

In ESD, after the margins of the lesion are visualized, the resection borders are marked using argon plasma coagulation (APC) or ESD knife [103]. This is followed by submucosal injection under the markings to create a cushion. An ESD knife is then used to perform a circumferential incision guided by the markings. Dissection is then performed with ESD knife and submucosal injection is repeatedly utilized to promote dissection within submucosal plane (Figure 2). After achieving dissection, the resection bed is closely inspected for signs of perforation or exposed vessels that may be at risk for bleeding.

A prior meta-analysis of 22 studies was published comparing outcomes of ESD and EMR for patients with superficial esophageal cancers including SCC, BE-associated neoplasia, and EAC [104]. ESD was associated with higher rates of en bloc resection, curative resection rate, and R0 resection, in addition to decreased rates of local recurrence. Subgroup analyses found that en bloc resection and curative resection rates were similar when lesion size was ≤10 mm, and local recurrence rates were similar when lesion size was ≤20 mm. There were more perforations in the ESD group, primarily the subgroup with ESCC, but the risk of bleeding and stricture were similar.

Overall, the findings of these studies support using ESD to treat lesions greater than 20 mm with higher risk for submucosal invasion and lesions with positive margin or prior incomplete resection [104,105,106].

## 7. Ablation Therapy

For patients who undergo ER for EAC, it is recommended that ablation therapy is utilized to eradicate residual BE with the goal of CEIM [15]. One prior study found an incidence of metachronous lesions of 36.7% in patients who underwent surveillance after ER without EET [107]. The higher risk for metachronous neoplasia in the residual BE segment of patients with esophageal adenocarcinoma is thought to be due to carcinogenic field effect, in which genetic alterations in tissue surrounding malignant tumors can predispose to cancer [108].

As discussed earlier, ablation therapy is also indicated for patients with BE with HGD given the high risk of progression with surveillance alone [15,56,57,58]. Furthermore, since a prior study has shown ablation therapy reduced the risk of progression of LGD, ablation can also be considered in confirmed LGD on the basis of shared decision making [55]. There are multiple types of ablative therapies that have been reported.

### 7.1. Radiofrequency Ablation

Radiofrequency ablation (RFA) utilizes high-power radiofrequency thermal energy bipolar electrodes to deliver heat that leads to coagulation of proteins and cell necrosis [109]. Furthermore, the consistent ablation depth of 0.5 mm using RFA allows for controlled and uniform ablation. Due to the limited depth of ablation, RFA should be performed only after nodular and raised lesions are resected.

Typically, circumferential ablation is performed using the Barrx^TM^ 360 Express ablation balloon (Medtronic; Sunnyvale, CA, USA). Following circumferential ablation or in cases in which circumferential BE is not present, focal ablations can be performed using the Barrx^TM^ RFA focal catheters or channel RFA through-the-scope catheter (Figure 3).

RFA demonstrated reduction of progression of BE-LGD and similar overall survival and EAC-related mortality in patients with HGD/intramucosal EAC when compared with esophagectomy [53,56]. Furthermore, a trial comparing this technique with stepwise radical ER showed that RFA led to CEIM in 96% of patients with fewer procedures (3 vs. 6) and lower risk of esophageal stenosis compared to ER [110]. Given the number of studies that support its efficacy, RFA is the most widely used ablation technique [93].

Rate of adverse events with RFA has been reported to be 8.8% in a prior meta-analysis [111]. This includes 5.6% of patients developing strictures, 1% having bleeding, and 0.6% developing perforation. Furthermore, the risk of all adverse events was significantly higher when RFA was performed with EMR (RR 4.4) compared to without EMR.

### 7.2. Photodynamic Therapy

Photodynamic therapy (PDT) is another method that has been studied for mucosal ablation in BE and esophageal cancer [112]. This employs the intravenous (IV) administration of agents to sensitize the esophageal mucosa to light. Light is then delivered to cause local injury, and the degree of tissue penetration depends on the agent and wavelength of light used. In a randomized multicenter trial including patients with BE with HGD, PDT led to elimination of HGD in 77% of patients within 5-year follow-up, which was found to be significantly higher compared to PPI alone [113]. Another trial demonstrated complete remission of dysplasia in 98% of patients with BE with LGD [114].

There are multiple limitations that have been described with this technique [112]. The lack of understanding of ideal dosage of light leads to risk of incomplete eradication of dysplasia with too little light and risk of necrosis and stricture formation with too much light. A study found that PDT was significantly less effective than APC in achieving macroscopic squamous re-epithelialization in BE [115]. Furthermore, the reported incidence of stricture formation was 16–29% in patients who underwent PDT for BE [116]. These findings have led to a decrease in use of this modality for treatment of esophageal neoplasia in favor of other ablative modalities [93].

### 7.3. Argon Plasma Coagulation

APC utilizes a jet of argon gas in a non-contact manner to induce superficial thermal effects to selectively destroy tissue [117]. A randomized trial found that APC led to a dysplasia clearance rate of 83.8% and BE clearance rate of 48.3%, which were similar to clearance rates using RFA [118]. However, limitations described with both APC and RFA include stricture formation and risk of recurrence and buried BE glands with neoplastic potential under neosquamous epithelium [118,119].

One newly developed technique to improve efficacy and safety of APC is hybrid-APC [120]. This technique first entails a lifting agent to create a visible cushion of the BE area, followed by ablation using APC until coagulation effect is visible (Figure 4) [121]. The initial debris formed by the first ablation is scraped off using a distal attachment cap, and second pass of additional injection and/or APC is performed to achieve the goal tissue effect. The proposed advantage of performing submucosal injection prior to APC is to provide an adequately aggressive ablative effect to the epithelium while minimizing risk of deeper injury [122].

A recent multicenter prospective trial of 154 patients found that hybrid-APC led to initial CEIM in 87.2% of patients and sustained CEIM in 70.8% with 4% overall stricture rate [123]. There have been no prospective randomized controlled trials to compare outcomes of hybrid-APC and RFA.

### 7.4. Cryotherapy

Cryotherapy is an ablation technique that involves a combination of freezing and thawing to induce tissue injury, and it has been utilized for treatment of various oncological conditions [124]. The administration of liquid nitrogen leads to a rapid cooling phase resulting in membrane disruption, protein denaturation, and cell dehydration [125]. This is followed by a slow thawing phase that ultimately leads to hypoxia and coagulation necrosis of targeted tissue.

There are two available types of cryotherapy: the truFreeze Spray Cryotherapy system (Steris; Dublin, Ireland) and the Cryoballoon Focal Ablation System (Pentax Medical; Montvale, NJ, USA). With the truFreeze system, liquid nitrogen is applied using a spray catheter, and a decompression orogastric tube is placed prior to treatment for gas venting to reduce risk of perforation [126]. Initial studies have shown that spray cryotherapy may lead to rates of eradication of dysplasia and BE similar to RFA with less postprocedural pain [127,128].

The Cryoballoon system involves the application of liquid nitrous oxide through a catheter with a compliant, transparent balloon tip of 3 cm in length. A prospective, multicenter trial found that use of Cryoballoon led to dysplasia clearance rate of 97% and BE clearance rate of 91% [129]. Cryoballoon and spray cryotherapy have also been shown in a meta-analysis to be effective in patients who did not initially respond to RFA, but further studies are required to validate this [130].

## 8. Endoscopic Palliative Therapies

With advanced and incurable esophageal cancer, a significant number of patients are affected by dysphagia and malnutrition [131]. There are an increasing number of therapeutic options available, including chemoradiation and brachytherapy, that have been shown to improve dysphagia and survival [132]. However, for patients with shorter life expectancy that require more rapid improvement in dysphagia or do not respond to chemoradiotherapy, endoscopic options are available for palliation.

### 8.1. Esophageal Stent Placement

Esophageal stent placement has become the most commonly used endoscopic therapy for palliation [133]. Due to the immediate and sustained relief it can provide, an esophageal stent is the preferred method for palliation of malignant dysphagia [134]. It is recommended that the underlying esophageal stenosis is thoroughly inspected via endoscopy to determine feasibility of stent placement [132]. Use of fluoroscopy can be considered to assist with choosing stent size and allow for localizing before, during, and after stent deployment [135]. A stent length should be chosen that will allow for the proximal end of stent to be at a minimum 2 cm above proximal tumor margin and the distal end to be at a minimum 2 cm below distal tumor margin.

Self-expandable metal stents (SEMS) are a widely accepted stent type used for esophageal cancer due to their flexibility and ability to exert self-expansive radial forces to reach maximum diameter [136]. At this time, partially or fully covered SEMS are recommended to minimize risk of tumor ingrowth and need for reintervention associated with uncovered stents [137]. Following selection of type and length of SEMS, typically a guidewire is inserted into the accessory channel of endoscope and across the stricture. This is followed by withdrawal of endoscope, passing the stent delivery system over the guidewire and stent deployment (Figure 5) [132].

Overall, technical success of endoscopic SEMS placement has been reported to be 97–99%, and rates of immediate (within 1 day) and delayed (at 4 weeks) improvement have been shown to be favorable [138,139,140]. It is noted that the rate of complications with stent placement have been reported to be 21–46%, which include stent migration, occlusion, reflux, aspiration, and pain [140,141,142]. Rare but catastrophic complications of tracheal or aortic fistulization have also been reported. Endoscopic suture fixation of the stent to the esophageal wall has been shown to potentially reduce the rates of migration [143]. Furthermore, stent occlusion due to hyperplastic tissue ingrowth can be managed endoscopically with methods such as placing another stent or performing APC [144,145].

### 8.2. Additional Palliative Therapies

Endoscopic dilation of malignant dysphagia can provide immediate improvement in patients with malignant dysphagia [146]. However, this technique is not commonly performed due to limitations including the incidence of recurrent dysphagia requiring repeated dilations and higher risk of perforation in malignant strictures [146,147].

In addition to its use for treatment of BE and early esophageal cancer, APC may have a role in the palliation of advanced esophageal cancer. A prior study found that 85% of patients had significant reduction in tumor size after APC, and 94% achieved improvement in dysphagia in 94% [148]. Furthermore, a randomized trial found that PDT in conjunction with APC led to a longer dysphagia-free period compared to APC alone [149].

Endoscopic cryotherapy has also been studied as a potential therapeutic option for palliation in patients with inoperable esophageal cancer (Figure 6). Prior retrospective studies have found that this technique led to improvement of dysphagia in 59–61% of patients [150,151]. While these findings support the potential efficacy of cryotherapy, it is noted that most patients required multiple endoscopic sessions and reported rates of strictures were 2–13%.

## 9. Esophageal Leaks, Fistulas, and Perforations

Esophageal leaks, fistulas, and perforations represent a challenging and life-threatening complication in esophageal cancer. Patients who undergo esophagectomy for cancer are especially at risk for this, as cervical anastomotic leaks occur in 10–20% of cases and thoracic anastomotic leaks occur in 5–10% of cases [152]. However, patients who undergo non-operative management may develop leaks including esophageal fistulas in setting of radiation therapy or disease progression itself [153,154]. Furthermore, acute perforation may be a potential complication of endoscopic therapies including dilation for malignant dysphagia or endoscopic resection for curative intent [155].

### 9.1. Initial Management of Esophageal Leaks

Prompt evaluation and management of leaks are essential to minimize complications that include mediastinitis, sepsis, multiorgan failure, and death [156]. To localize the leak and determine the extent, patients should first undergo an esophagram using Gastrografin, which is water-soluble and preferred as initial contrast agent over barium due to lower risk of mediastinal and pleural inflammation [157]. However, barium esophagram is more sensitive in detecting esophageal leaks and is recommended if Gastrografin esophagram is negative and suspicion remains high [158]. Furthermore, obtaining a CT would also be useful to detect air leaks and fluid collections [159].

Once an esophageal leak is diagnosed, patients should be kept nil per os (NPO), resuscitated with IV fluids, and started on an IV PPI. To further minimize the complications of infection, administration of IV antibiotics, antifungals, and nasogastric tube is recommended [160,161]. Initial source control is usually achieved with percutaneous or operative tube placement to provide drainage.

### 9.2. Endoscopic Management of Leaks, Fistulas, and Perforations

Surgery has generally been the intervention utilized for repair of esophageal leaks, fistulas, and perforations. With the evolving therapeutic applications of endoscopy, there may be an increasing role of endoluminal therapy that may prevent the need for general anesthesia, thoracotomy, and dissection [152]. The decision to pursue endoluminal therapy should take into account the technical and anatomic aspects, and this should be considered following a multidisciplinary discussion that includes thoracic surgery, interventional radiology, and gastroenterology [162].

#### 9.2.1. Endoscopic Placement of Clips

Endoscopic placement of clips is another available option to treat esophageal leaks. Conventional through-the-scope clips, due to their smaller size, have previously been noted to have difficulty in approximating defect margins for closure [163]. The introduction of over-the-scope clips (OTSCs) has helped overcome these limitations through its larger clip area and greater compression force [164]. An applicator cap with the mounted clip is attached to the end of the endoscope. Once the margins of the tissue defect are grasped with the cap via suction, the clip is deployed and provides a full thickness closure of the wall.

A prior study of 76 patients with anastomotic leaks found that OTSC led to closure and clinical success in 83% of patients [165]. Furthermore, another retrospective study demonstrated that success with OTSC was significantly higher when used for primary closer of defects as opposed to rescue therapy [166]. At this time, application of OTSC is recommended for defects up to 1–1.5 cm [167,168]. The mucosa should also be free of edema or ulceration that may prevent clip retention and seal [152].

#### 9.2.2. Endoscopic Suturing

Endoscopic suturing is an emerging minimally invasive technique with applications in endoscopic bariatric therapy, stent fixation, and defect closure [168]. The Apollo Overstitch system (Apollo Endosurgery Inc.; Austin, TX, USA) is a single-operator platform that allows for full-thickness suturing [169]. With a tissue helix that allows for tissue grasping and retraction to promote approximation, the Overstitch allows for placement of interrupted or continuous sutures without needing to remove the device. In multiple case series, this technique has been successful in closing acute perforations of the esophagus due to Boerhaave syndrome and iatrogenic causes [169]. For management of gastrointestinal fistulas, a retrospective study found that although immediate closure was achieved in 100% of cases, only 22.4% maintained closure at 12 months [170]. This highlights the challenge of closing fistulas due to the epithelialized tract and edges, which may benefit from combining suturing with adjunct therapies including APC, clips, and placement of a stent to divert enteral contents. Currently, endoscopic suturing is a potential therapeutic option of esophageal leaks that requires studies with long term follow up for this technique.

#### 9.2.3. Tissue Sealants

Endoscopic application of tissue sealants represents another therapeutic option for treatment of esophageal leaks. Two types of sealants, fibrin glues and cyanoacrylates, are used in gastrointestinal surgery for the prevention and management of anastomotic leaks [171]. It is believed that these can promote tight approximation of anastomosis and wound healing with minimal fibrosis [171,172]. Multiple prior studies found that intraluminal and submucosal fibrin glue injection led to successful closure of anastomotic leaks, and the addition of a Vicryl plug led to more rapid closure and earlier introduction of oral nutrition [173,174]. However, the evidence behind the use of tissue adhesives is limited to case series and case–control studies. Future comparative studies of this technique would be informative.

#### 9.2.4. Esophageal Stent Placement

Esophageal leaks can also be managed with endoscopic stent placement, which is thought to divert secretions away from the site of dehiscence to promote healing. For cases of dehiscence ≥ 30% of the esophageal circumference, stent placement is preferred compared to other endoscopic modalities [152]. There are multiple types of stents that can be utilized, including fully covered SEMS, partially-covered SEMS, and self-expandable plastic stents. Prior studies have reported leaving stents in place for varying periods ranging from 14 to 256 days [175,176].

Overall, endoscopic stent placement has a technical success rate of 91% and complete leak/perforation healing rate of 81% for esophageal anastomotic leaks and perforations [177]. As discussed earlier, stent migration is a potential adverse event, especially with fully covered SEMS, and the risk can be mitigated with suture or clip fixation of the stent to esophageal wall [143]. A prior comparative analysis found that stent failure was more frequent in patients with leaks in proximal cervical esophagus, leaks traversing the gastroesophageal junction, esophageal injury longer than 6 cm, and anastomotic leak associated with a more distal conduit leak [178].

#### 9.2.5. Endoscopic Catheter Drainage and Debridement

Drainage of underlying mediastinal abscesses and removal of necrotic material are necessary prior to closure of an esophageal leak. A protocol for endoscopic treatment of paraesophageal abscesses after esophageal perforation or postoperative leakage has been described in Germany [179]. The mediastinal abscess cavity is accessed either through creating a tract by using linear EUS or directly entering the cavity via pre-existing perforation site with aid of a guidewire. Under fluoroscopic guidance, a 7Fr catheter is introduced over the guidewire and pus is aspirated. If a significant amount of pus is drained, then a plastic stent is left in the tract to assist with drainage. If simple drainage of pus is not achieved, the cavity is lavaged with sterile saline under endoscopic visualization, and necrotic material and debris are removed by endoscopic retrieval devices. This process of endoscopic lavage and debridement is repeated daily until no pus or debris are present. In this study, all 15 patients had successful debridement through a median of five daily sessions with improvement in clinical parameters. Overall, these findings demonstrate endoscopic drainage and debridement of mediastinal abscesses may be a feasible alternative to surgical intervention.

#### 9.2.6. Esophageal Vacuum Therapy

Endoscopic vacuum therapy (EVT) was first reported in 2008 and is an approach that has been studied in management of esophageal leaks. After the defect is assessed endoscopically, a vacuum sponge is either placed into the cavity or completely over the defect while remaining in the esophageal lumen [164]. Continuous suction is then applied through a nasogastric tube connected to the vacuum sponge. Through negative pressure therapy, the mechanisms behind the therapeutic effect include apposition of wound edges, wound drainage, formation of granulation tissue, neovascularization, and diversion of secretions from the site of healing [180]. It is recommended that the vacuum sponge is exchanged to maintain suction forces and allow for easier removal due to less granulation tissue infiltration.

Prior studies have reported closure rates between 60 and 100% in addition to high rates of success in rapid control of sepsis [180]. With mean healing times ranging from 12 to 36 days, potential limitations include the length of treatment period and number of exchanges required [164]. Furthermore, barriers to success with EVT include large multiloculated collections and proximity of visible large blood vessels that may be at risk of bleeding with negative pressure therapy [180]. Ultimately, the decision to pursue any endoscopic intervention for esophageal leak should be made as part of a multidisciplinary approach.

## 10. Conclusions

Esophageal cancer has remained one of the leading causes of cancer-related mortality worldwide. Previously, endoscopy has primarily served to aid in diagnosis of this condition. With advances in diagnostics and increasing knowledge regarding precursor lesions such as Barrett’s esophagus, there is a growing role of endoscopy for screening and surveillance. In addition, emerging endoscopic techniques have increased the therapeutic capabilities for curative intent, palliation, and management of complications. As new technologies are developed and explored, the applications of endoscopy for management of esophageal cancer should continue to expand.

## Figures and Tables

**Figure 1 cancers-14-03583-f001:**
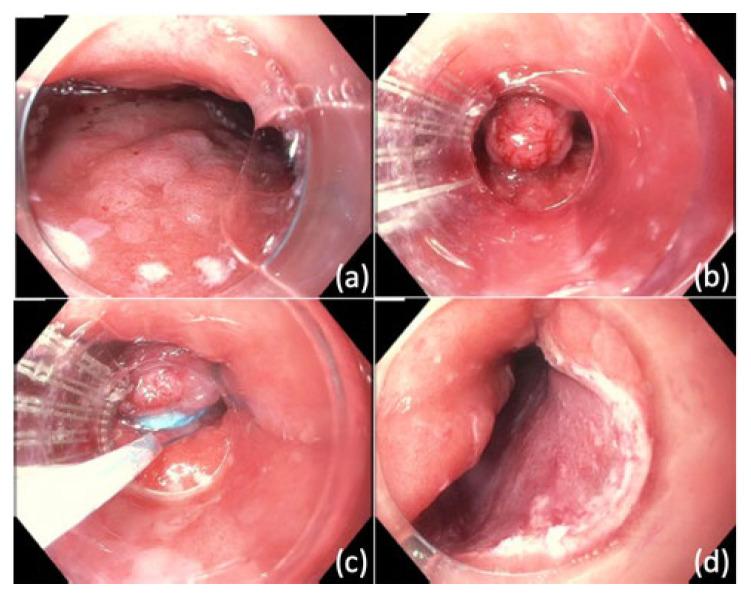
Ligation-assisted endoscopic mucosal resection (EMR). (**a**) Nodular mucosa representing high grade dysplasia in Barrett’s esophagus (**b**) Band deployed to create pseudopolyp (**c**) EMR performed using hot snare (**d**) Site of resection after ligation-assisted EMR.

**Figure 2 cancers-14-03583-f002:**
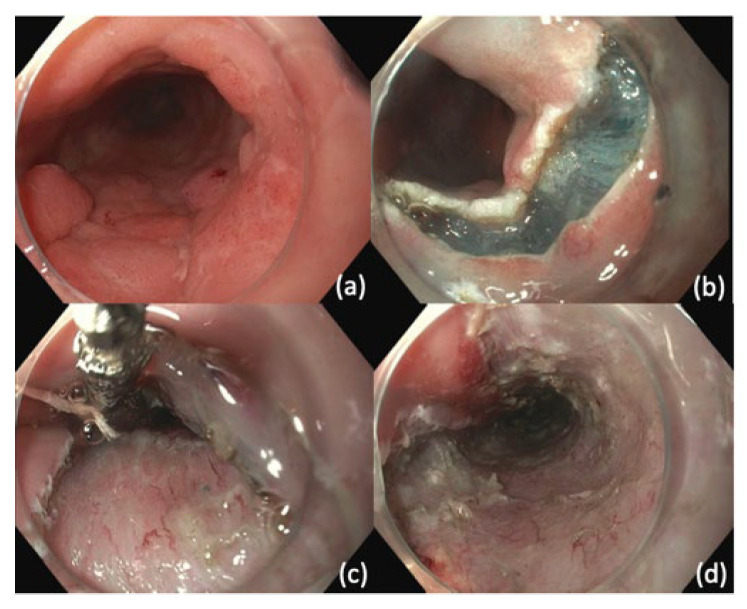
Endoscopic Submucosal Dissection (ESD). (**a**) Nodular mass lesion representing focally invasive adenocarcinoma (**b**) Distal mucosa incision performed after submucosal injection (**c**) Tunnel creation by ESD (**d**) Defect after completion of en bloc resection.

**Figure 3 cancers-14-03583-f003:**
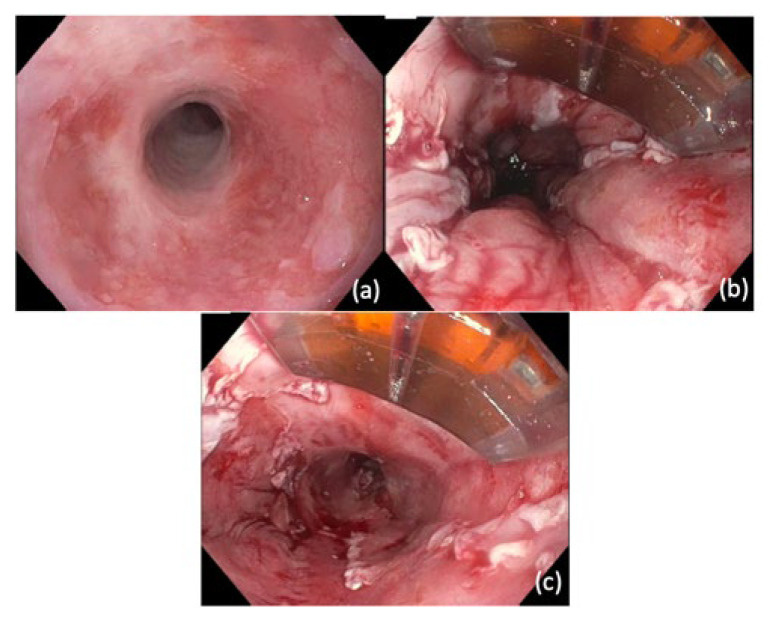
Radiofrequency ablation (RFA). (**a**) Evidence of Barrett’s esophagus and stenosis associated with prior endoscopic mucosal resection (**b**) RFA performed using Barrx^TM^ Halo Ultra Long Catheter (**c**) Segment of Barrett’s esophagus after RFA.

**Figure 4 cancers-14-03583-f004:**
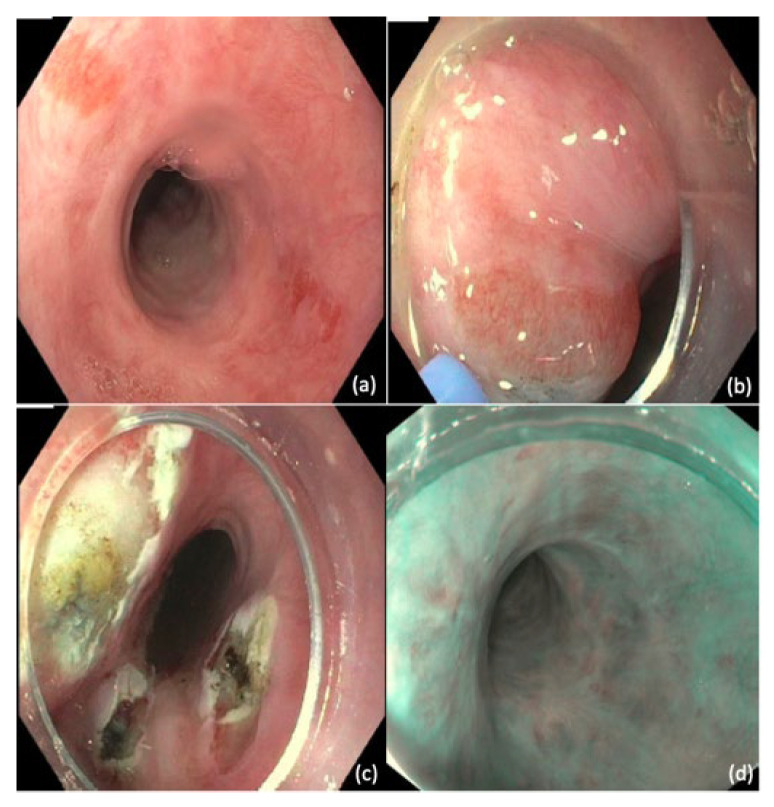
Hybrid argon plasma coagulation (Hybrid-APC). (**a**) Focal islands of Barrett’s esophagus (**b**) Submucosal lift created using a hybrid-APC catheter (**c**) Focal islands after treatment with hybrid-APC (**d**) Neosquamous epithelium identified on narrow band imaging at follow-up endoscopy.

**Figure 5 cancers-14-03583-f005:**
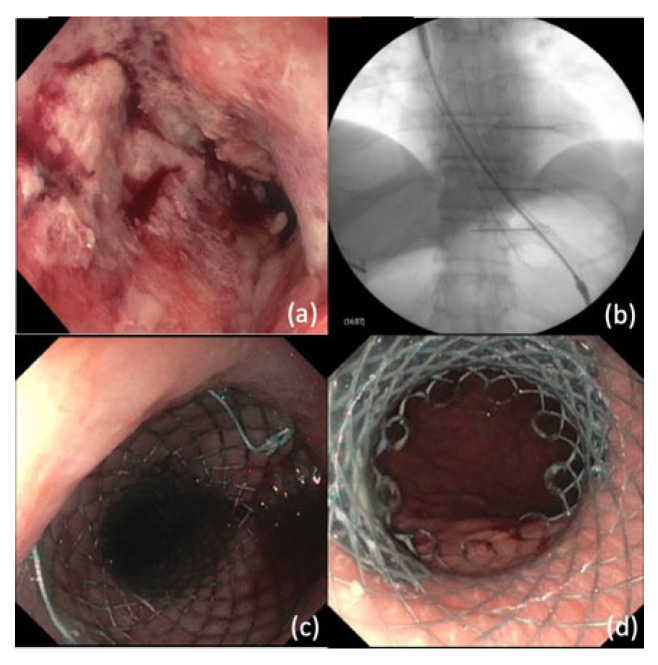
Esophageal stent placement. (**a**) Moderate luminal narrowing from squamous cell carcinoma (**b**) Placement of partially-covered metal stent over a wire under fluoroscopic guidance (**c**) Proximal end of partially-covered metal stent (**d**) Distal end of partially-covered metal stent.

**Figure 6 cancers-14-03583-f006:**
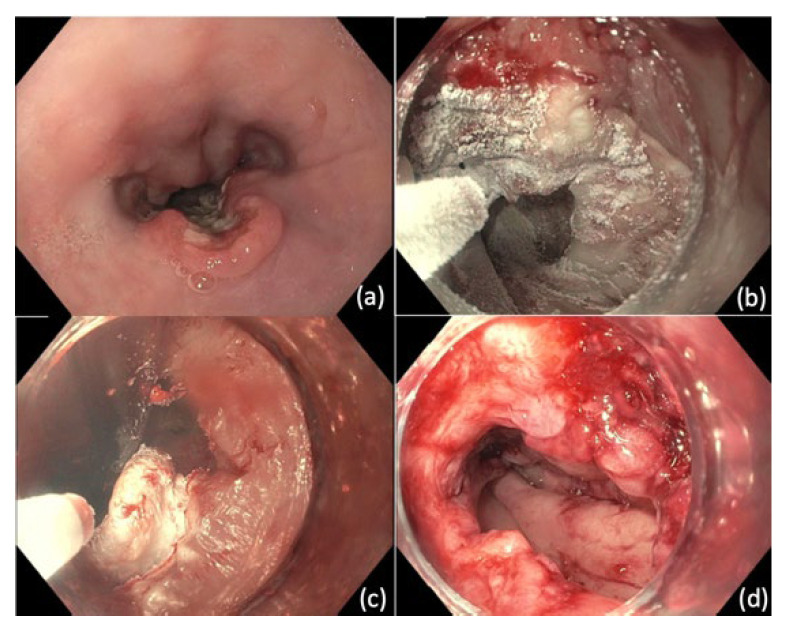
Palliative endoscopic cryotherapy. (**a**) Ulcerated and friable mass lesion representing squamous cell carcinoma (**b**) Application of liquid nitrogen using a spray catheter (**c**) Treatment of another area of nodularity (**d**) Site of cancer after cryotherapy.

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
