# Peer review of "Endoscopic Management of Esophageal Cancer"

_cancers, 2022, doi:10.3390/cancers14153583_

Round 1

Reviewer 1 Report

This paper describes the endoscopic management of esophageal cancer. I enjoyed reading this paper and I think the authors discussed precisely and their review will provide useful information for gastroenterologists. However, I have some comments.

Comments

1.    In the chapter “2. Evaluation of Barrett’s Esophagus”, the author mentioned not only Barrett’s cancer but also esophageal squamous cell cancer (ESCC). I recommend they should make other chapter for squamous cell cancer.

2.    Narrow band imaging is also useful for T staging of ESCC. Please add in the chapter “4. Staging of Esophageal Cancer”.

3.    I thought the authors focused on managements for esophageal leak in the chapter 7.2 and 7.3. I suppose the name of chapters should be changed to make readers understand.

4.    The chapter 7.1 are duplicated.

Reviewer 2 Report

Dear authors,

I was pleased to review the article “Endoscopic Management of Esophageal Cancer”. The methodology used by the authors is appropriate for the purpose of the study and conclusions are narrowly linked to available evidence.

This review article contains complex and complete information about the endoscopic management of esophageal cancer but, in some paragraphs, the information goes beyond the subject of interest analyzed, namely esophageal cancer and endoscopy in esophageal cancer.

Regarding this, the paragraph 2. Evaluation of Barrett’s Esophagus  contains many details regarding the involvement of histopathology. I recommend that these pathological details be reduced. The paragraph 7.2 . Medical Management contains details that are not the subject of this article.

There are 2 paragraphs that are numbered the same: 7.1. Esophageal Stent Placement and 7.1. Additional Palliative Therapies

The Manuscript may benefit from some other minor revisions, as suggested below:

-        the figures 2, 3, 4 and 5 presented in the article contain many details that I suggest to be included in the text, and in the figure to be mentioned only references

-        the figures 1 and 6 - as these are images taken from other articles, the reference needs to be mentioned

-        I suggest a revision of the References, because references should not include the month in which an article was published

In conclusion, I agree that this manuscript should be accepted for publication, after these revisions. 
